# Effectiveness of Artificially Synthesized Granitic Residual Soil-Supported Nano Zero-Valent Iron (Gr-nZVI) as Effective Heavy Metal Contaminant Adsorbent

Nur 'Aishah Zarime [1],*[ID], Badariah Solemon [1],*, Wan Zuhairi Wan Yaacob [2], Habibah Jamil [2], Rohayu Che Omar [1] and Adeleke Abdulrahman Oyekanmi [1]

[1] Institute of Energy Infrastructure (IEI), Universiti Tenaga Nasional (UNITEN), Putrajaya Campus, Jalan IKRAM-UNITEN, Kajang 43000, Selangor, Malaysia; rohayu@uniten.edu.my (R.C.O.); abdulkan2000@yahoo.com (A.A.O.)

[2] Department of Earth Sciences and Environmental, Faculty of Science and Technology, Universiti Kebangsaan Malaysia, Bangi 43600, Selangor, Malaysia; yaacobzw@ukm.edu.my (W.Z.W.Y.); bib@ukm.edu.my (H.J.)

* Correspondence: aishahz@uniten.edu.my (N.'A.Z.); badariah@uniten.edu.my (B.S.)

**Abstract:** Supported nano zero-valent iron is receiving great attention nowadays due to its effectiveness in treating heavy metal pollutants. Therefore, this study aimed to investigate the effectiveness of granitic residual soil-supported nano zero-valent iron (Gr-nZVI) for the removal of the heavy metals $Pb^{2+}$, $Cu^{2+}$, $Co^{2+}$, $Cd^{2+}$ $Ni^{2+}$ and $Zn^{2+}$ in mixture solutions under different experimental conditions of batch equilibrium tests. In this study, Gr-nZVI was successfully synthesized by using the chemical reduction of Ferric Chloride Hexahydrate ($FeCl_3.6H_2O$) and Sodium Borohydride ($NaBH_4$). The physical and chemical properties, morphology and mineralogy of all adsorbents were characterized by the Braeuer–Emmett–Teller (BET) method, cation exchange capacity (CEC), X-ray fluorescence (XRF), scanning electron microscopy (SEM), field-emission scanning electron microscopy (FESEM), X-ray diffraction (XRD) and Fourier transform infrared spectroscopy (FTIR). Isotherm, kinetic and diffusion model analyses were conducted to fit the experimental data. The results show rapid adsorption within 5 min in the initial adsorption stage for $Pb^{2+}$ on nZVI ($q_{e.Pb}$ = 17.89 mg/g) and Gr-nZVI ($q_{e.Pb}$ = 15.29 mg/g). nZVI and Gr-nZVI also showed no significant effects on pH and temperature, serving as a good example of an energy-efficient process. The isotherm data fitted better to the Langmuir model and the pseudo-second-order kinetic model for the adsorption of all of the heavy metals. The diffusion models revealed that adsorption was not the only rate-limiting step. In this study, nZVI compared to Gr-nZVI and Gr demonstrated superior adsorption capacity for the heavy metal adsorption selectivity. Hence, these materials can be utilized as alternative energy-efficient adsorbents for the adsorption of metal ions from wastewater.

**Keywords:** adsorption; heavy metals; supported nano zero-valent iron; granitic residual soil; energy-efficient

## 1. Introduction

The presence of heavy metals in the environment at concentrations beyond acceptable thresholds poses a serious threat to human health upon exposure and has harmful effects on biological systems [1]. High concentrations of heavy metals in the surface water also disturb the natural balance of the aquatic ecosystem [2]. The discharge of effluents containing heavy metals into the receiving water without effective treatment has become a major source of water pollution. Consequently, the availability of pristine water for industrial and domestic utilization has been significantly reduced over the years [3,4]. To ensure the preservation of surface water for its effective utilization and environmental preservation, several treatment methods have been proposed. Some of the methods include chemical precipitation, ion exchange, filtration, membrane separation and adsorption [1,2,5]. However, adsorption

has a comparative advantage over other treatment techniques due to its simple operation, its high efficiency in the removal of heavy metals, its low energy requirement, its economic feasibility, and the substantial options for adsorbent materials [6].

Due to their numerous beneficial applications in a variety of scientific fields, nanomaterials have garnered considerable attention [6]. Nanoparticles are used to a large extent because of their intrinsic capabilities, great efficiency in the reduction in reactions, high reactivity due to higher surface area, high mobility and high filtration efficiency, especially for remediation under different environmental conditions [7]. Because nanoparticles are unstable, it may be necessary to modify their surfaces. Additionally, the modification of each nanoparticle depends on the contaminant that is being targeted. This requires the customization of nanoparticles based on the characteristics of the wastewater [8]. The use of organic adsorbents for the removal of heavy metals in solutions [9] is well documented in the literature [10–12], including the development of composites [13,14]. Although numerous studies have demonstrated the high efficiency of various nanoparticles, the use of nanotechnology in wastewater treatment is still in its early stage due to the high operating costs of this field of research. An economic evaluation was conducted by [15] to determine whether a project was feasible. The parameter used to perform this economic evaluation is the cost of raw materials, utilities, sales, salaries, and taxes, which are varied under varying circumstances. Previous researchers [15] also calculated the cost of using magnesium oxide nanoparticles, and the total production was estimated to be 19,055 USD/year.

Composite nanoparticles can be used to stabilize nZVI particles. This is due to nanoparticles having a strong tendency to agglomerate into larger particles, resulting in reduced reactivity. According to [9], the use of nanoparticles for remediation can cause the breakdown of nanomaterials, resulting in environmental contamination. To protect people and the environment from these potential risks, these novel materials should not be widely used until such losses can be prevented. Granite is a magmatic rock produced on a large scale via extraction. It is estimated that about 50% of the extracted granite corresponds to dry waste generation. This waste is accumulated in the environment over time and, as a result, generates a serious environmental impact. In addition, granite wastes have toxic effects on workers in the mining industry and can be a source of respiratory diseases such as cardiorespiratory disease [16]. However, this waste residue can be recycled and reused as a low-cost viable alternative adsorbent.

In this study, granitic residual soil was used as a composite material to synthesize granitic residual soil-supported nano zero-valent iron (Gr-nZVI) for heavy metal treatment. Conventionally, clay has always been processed as a composite. However, in the case of the utilization of Gr, there is a higher percentage of sand compared to clay, which implies that there is a high tendency for poorer surface properties and adsorption capacity. Therefore, this study investigated the application of Gr-nZVI for the adsorption of heavy metal ions from an aqueous solution. The characteristics, adsorption behaviors, effects of process factors and diffusion mechanisms, including the empirical isotherm and kinetics, were evaluated to determine the sustainability of the synthesized adsorbent for heavy metal adsorption. The synthesis of Gr-nZVI has never been reported in the literature. This research innovation could help in controlling heavy metals and improve their adsorption capacity in Gr soils, as they are present in landfill areas and also used as industry-based materials in Malaysia.

## 2. Materials and Methods

### 2.1. Materials and Chemicals

In this study, Gr, which was utilized as the precursor material, was collected in Broga, Selangor, Malaysia. The in situ technique was applied during the sampling of Gr. The soil sample was taken with a shovel at a depth of approximately 5 cm from the surface. The sample was then dried at room temperature in the laboratory and sieved using a 63 μm sieve size prior to the investigation. To synthesize the nanocomposite, Ferric Chloride Tetrahydrate ($FeCl_3.6H_2O$; Acros organics, 99 +%), Sodium Borohydride ($NaBH_4$; Acros

organics, 98 +%) and Ethanol ($C_2H_6O$; Fisher Scientific, 99.4%) were supplied by R&M chemicals located in Semenyih, Selangor Malaysia. All chemicals used in the study were of analytical grade unless stated otherwise. The simulated water used for the adsorption study was prepared using chemical reagents, including $Pb(NO_3)_2$, $Cu(NO_3)_2$, $Co(NO_3)_2$, $Cd(NO_3)_2$, $Ni(NO_3)_2$ and $Zn(NO_3)_2$. These chemicals were supplied by Orc Chemicals located in Shah Alam, Selangor Malaysia, and were in a solid state with high purity (99.9 wt.%) unless otherwise stated.

### 2.2. Synthesis of nZVI and Gr-nZVI Nanocomposites

nZVI and Gr-nZVI were prepared through a modified chemical reduction method [17]. $FeCl_3.6H_2O$ solutions were prepared by dissolving 4.38 g of $FeCl_3.6H_2O$ in an ethanol/water solution (50 mL, 4:1 *v/v*). Gr was added to the $FeCl_3.6H_2O$ solution, and the mixture was stirred in an ultrasonic shaker for 30 min (this procedure was not conducted for bare-nZVI). About 6.091 g of $NaBH_4$ was added to a conical flask containing 100 mL of deionized water. The $NaBH_4$ solution was slowly added to composite $Gr-FeCl_3.6H_2O$ mixture solutions. The solid black nanocomposite was gradually produced. Subsequently, the mixture was stirred for 20 min and washed three times using ethanol. Finally, the solid black nanocomposite obtained was oven-dried for 12 h at 50 °C. The synthesized nanocomposite was stored in an airtight vessel until used for the adsorption study.

### 2.3. Characterization of the Adsorbents

The physical properties of the adsorbent samples were characterized by Brunauer–Emmett–Teller (BET). Through this test, the surface area, pore size and pore volume of the adsorbent samples were determined. The analysis was conducted using the nitrogen adsorption procedure with Micromeritics ASAP 2020 model (Micromeritics Instrument Corporation, Norcross, GA, USA) equipment. The adsorbent samples were degassed at 350 °C for 8 h under inert nitrogen gas flow for the removal of adsorbed moisture and organic compounds, which are likely to block the pores. The chemical composition of the adsorbents was measured by using cation exchange capacity (CEC), which was determined according to ASTM D4319 methods. The surface morphology of the materials was recorded by scanning electron microscopy (SEM) and field-emission scanning electron microscopy (FESEM). SEM was performed on a Q150RS Quorum instrument, with micrographs obtained at ×5000 magnification. FESEM images were obtained using a Zeiss (Model Merlin/Merlin Compact/Supra 55VP) instrument, operating at 1.7 nm @ 1.0 kV and 4 nm @ 0.1 kV. The device was fully controlled by a computer using Zeiss SmartSEM$^{TM}$ software. The mineralogy of adsorbents was investigated by X-ray diffraction (XRD) and Fourier transform infrared spectroscopy (FTIR). XRD analysis was performed using a D8-Advance (Bruker AXS Co., Ltd., Billerica, MA, USA). The FTIR analysis was conducted to obtain the functional properties of the adsorbents using the Nicolet 6700 Thermo Nicolet, Nexus 670 Spectrometer (Thermo Electron Scientific Instruments Corporation, Madison, WI, USA) with bands obtained at a resolution of 4 $cm^{-1}$.

### 2.4. Adsorption Study

2.4.1. Batch Equilibrium Test

The experimental isotherm data were obtained through batch adsorption tests for heavy metals and were calculated according to standard methods [18]. Batch equilibrium tests were also conducted according to the standard method from USEPA (1992) [18] to determine the effect of process variables at defined intervals under different operational conditions, which include the adsorbent dosage (0.03–1 g), initial concentration (5–250 mg/L), pH (2–12), contact time (5–360 min) and temperature (30–60 °C). The concentrations of heavy metals absorbed by the absorbent ($q_e$) were calculated using the following formula:

$$q_e = \frac{\left(C_o - C_f\right)V}{M} \tag{1}$$

where $C_o$ and $C_e$ are the initial concentration and equilibrium concentration (mg/L), respectively, $V$ is the volume of solution added (mL), and $M$ is the mass of air-dried material (g).

### 2.4.2. Adsorption Isotherm

Adsorption isotherms were studied to understand the mechanism of heavy metal adsorption onto the adsorbent materials and also to determine the maximum adsorption capacity of the adsorbent materials [19]. Equilibrium isotherm data were fitted to empirical models, as shown in Table S1. The partition coefficient, $K_d$, was also used to estimate the adsorption of dissolved contaminants in contact with adsorbent materials [20]. $K_d$ was obtained from the relationship of $q_e$ versus $C_e$ by referring to the formula below [21].

$$q_e = K_d \times C_e \qquad (2)$$

The experimental results were examined using equilibrium isotherm models, namely, Langmuir isotherm and Freundlich isotherm equations, to assess the most suitable equilibrium model for heavy metal adsorption onto the synthesized adsorbent materials. The interpretation of the Langmuir isotherm is described in Table S1. The Langmuir isotherm is the simplest model and is valid for monolayer adsorption on a surface containing a limited number of sites, predicting a homogeneous distribution of adsorption energies and indicating no substantial interference between the adsorbates [19]. The Freundlich isotherm, also known as a variational sorption model, suggested the heterogeneous energy distribution of active sites, followed by encounters between adsorbed molecules [19].

### 2.4.3. Adsorption Kinetics

To understand the kinetics of the adsorption process, the experimental data were fitted to two main types of models, namely, reaction-based models and diffusion-based models. The reaction models consist of the pseudo-first-order and pseudo-second-order kinetic models. The diffusion-based models consist of two models, which are the external diffusion model (known as the interparticle diffusion model) and the internal diffusion model (known as the intraparticle diffusion model) [22]. Table S1 also indicates the details and equations for all kinetic models used in the study.

## 3. Results and Discussion

### 3.1. Material Characterization

The surface characteristics of the adsorbents obtained using the BET analysis shown in Table S2 indicated that the surface areas of nZVI, Gr and Gr-nZVI were 1.5724 m$^2$/g, 14.1724 m$^2$/g and 6.7619 m$^2$/g, respectively. From the results, it was revealed that Gr exhibited the highest surface area, followed by Gr-nZVI and nZVI. This trend also agrees with the values of pore volume, where Gr, Gr-nZVI and nZVI show values of 0.0484 cm$^3$/g, 0.0298 cm$^3$/g and 0.0042 cm$^3$/g, respectively. The lower values of the BET surface area and pore volume in Gr-nZVI clearly imply that the pores originated from voids between the particles and were filled with nZVI particles. The pore size of Gr-nZVI was also found to have the highest value (176.0732 Å) compared to those of other adsorbent materials. Previous researchers suggested that the larger the pore size, the higher the activation rate [23].

Table S3 shows the results of CEC. nZVI demonstrated the highest CEC value (4.03 ± 5.57 meq/ 100 g), followed by Gr (2.70 ± 9.03 meq/100 g) and Gr-nZVI (1.68 ± 7.24 meq/100 g). From the results, Gr as the composite material did not increase the CEC value of Gr-nZVI. Previous studies stated that soil with a high soil buffering capacity would cause the CEC to change [24]. However, the Gr used in this study was categorized as having a low soil buffering capacity based on its CEC [25]. Similarly, the increase in CEC could be influenced by many factors, such as soil pH, soil texture and organic matter content [26]. Therefore, an increase in CEC could be achieved by adjusting different operational factors. The chemical compositions of all samples were determined by XRF, and the results are presented in Table S4. The results revealed that iron ($Fe_2O_3$) content was higher in nZVI and Gr-nZVI by

equivalent percentages of 54.99% and 30.81%, respectively, while Gr had only 4.08% $Fe_2O_3$ content. The increased iron content in nanoparticles indicates that $Fe^0$ was successfully synthesized using $NaBH_4$ and $FeCl_2.6H_2O$ in both nZVI and Gr-nZVI.

SEM and FESEM were used to describe the morphologies of Gr and Gr-nZVI before and after treatment, as illustrated in Figure S1. As reported in [27], the SEM image of Gr (Figure S1a) only indicated its microstructure, which was mainly composed of sheets and plate structures of kaolinite. Through the FESEM image, the effect of heavy metal treatment on Gr was determined. The FESEM image depicted in Figure S1f shows that the clay sheets increased. Gr also contains a mixture of halloysite, quartz and kaolinite, which form dense aggregate structures [28]. According to [29], an increased heavy metal concentration also induces kaolinite particle aggregation. nZVI particles could be observed in nZVI (Figure S1b,c) and Gr-nZVI (Figure S1d,e). nZVI shows spherical shapes, with sizes ranging from 39.60 to 71.46 nm (nZVI) and 54.75 to 71.46 nm (Gr-nZVI). It was indicated that nZVI particles were aggregated into chain-like structures [30,31]. Kaolinite minerals also were identified in Gr-nZVI. Previous studies have stated that kaolinite has a high resistance to chemical damage if a high overburden pressure is applied [32,33]. Figure S1g shows that the nZVI structure was degraded after the treatment of heavy metals, and the nZVI particles had an uneven form and were difficult to differentiate from one another. These results correspond to the study by [34]. The nZVI particles remain aggregated into chains due to their magnetic properties and tendency to remain in the most endothermically favorable state [34]. Figure S1h depicts the Gr-nZVI surface morphology. The nZVI particles in Gr-nZVI were degraded, and according to [34], this may be associated with swelling black sheer. The swelling black sheer could have formed as a result of insufficient iron loading.

The XRD patterns for all samples are illustrated in Figure S2. As mentioned in [27], Gr before treatment, Gr(B), and after treatment, Gr(A), consists of kaolinite, which was identified from the peaks at $2\theta = 12.4°$, $24.9°$ and $36°$. Illite and halloysite minerals were also identified at peaks of $18.4°$ and $19.2°$, respectively. The differences between Gr(B) and Gr(A) are the intensity of minerals existing in the samples. The intensities of quartz, halloysite and kaolinite in Gr(B) are higher compared to the intensities of these minerals in Gr(A). For nZVI and Gr-nZVI, the apparent peaks at $2\theta = 44.9°$ indicated the presence of zero-valent iron for both samples [35]. In Gr-nZVI, the intensities of clay minerals such as kaolinite and halloysite were reduced or absent during the synthesis of the nanocomposite. The study by [34] stated that the compositional disturbance brought on by the intercalation of iron can make supporting minerals such as kaolinite and halloysite lose their reflection peaks after loading with iron. However, instead of disappearing, the quartz reflection peak was reduced. After the treatment, nZVI (A) exhibits the presence of a low-intensity Fe peak in comparison to nZVI (B). For nZVI (A), XRD patterns show the presence of only the $Fe^0$ peak, while for Gr-nZVI (A), the Fe peak is almost absent, and the intensity of kaolinite is lower compared to Gr-nZVI (B). Based on the study by [36], a reduced diffraction peak suggests that the internal pores of the adsorbent material largely collapsed since the acid or/and alkali pretreatment led to its disordered shape and lower crystallization.

The functional properties of nZVI, Gr and Gr-nZVI from the FTIR spectra obtained before the treatment are shown in Figure S3. The adsorption bands at 3960 $cm^{-1}$ and 3620 $cm^{-1}$ for Gr and Gr-nZVI (before and after treatment) samples correspond to outer and inner hydroxyl groups [37,38]. The OH groups on the outer or inner surfaces of the octahedral sheets form weak hydrogen bonds with the oxygens of the subsequent tetrahedral layer and vibrate in a phase-symmetric manner to produce a strong band around 3694 $cm^{-1}$. However, the strong band near 3620 $cm^{-1}$ is caused by the stretching vibrations of the "inner OH groups," which are located between the tetrahedral and octahedral sheets. After exposure to heavy metal treatment, the hydroxyl group interacts via metal–oxygen interactions, reflected in the band shift to a lower wavenumber, indicating the hydroxyl group's involvement in metal binding [39]. Garci et al. (2014) [40] also claimed that the O-H stretching vibration and -OH bending vibration of adsorbed water molecules are responsible for the appearance of a broad peak in the range between 3500 $cm^{-1}$ and

3200 cm$^{-1}$. The band at 3342 cm$^{-1}$ for nZVI and Gr-nZVI samples (before treatment) is attributed to OH stretching vibration and the one at 1640 cm$^{-1}$ is attributed to the OH bending vibration of surface-adsorbed water, indicating the presence of a ferrioxyhydroxide (FeOOH) layer on Fe$^0$ nanoparticles [41]. The study by [42] also supported that the peak at 1638 cm$^{-1}$ was triggered by the physical adsorption of certain water molecules on the surface of nZVI. After the heavy metal treatment, the vibration peaks of nZVI and Gr-nZVI decreased due to nanoparticle adsorption.

For nZVI and Gr-nZVI (before treatment), the adsorption bands at 1424 cm$^{-1}$ (nZVI) and 1410 cm$^{-1}$ showed that the nanocomposite materials had been successfully synthesized [43]. The adsorption band at 1340 cm$^{-1}$ occurred due to the presence of ethanol during the synthesis of the nanocomposite materials. According to [42,44], the 1415 cm$^{-1}$ band for nZVI and Gr-nZVI (before treatment) is caused by the asymmetric stretching vibration of -COO- and carboxylic ions. Both adsorbent materials also showed an adsorption band at 1252 cm$^{-1}$, which could be due to the stretching vibration of conjugated carbonyl groups. However, the intensity of Gr-nZVI at these bands was reduced due to the reduction of C=O groups. Thus, this reduction is expected to increase the reactivity of the synthesized Gr-nZVI [43]. The adsorption band at 990 cm$^{-1}$ for Gr, nZVI and Gr-nZVI (before treatment) appears sharp and steep and corresponds to the disruption of the Si-O bond [45]. The adsorption band at 750 cm$^{-1}$ for Gr corresponds to the inner surface vibration of Al-O-Si and is associated with the Si-O bond of the quartz mineral [37]. The FTIR results after treatment show that the nZVI band becomes wider and broader, and the intensity has been reduced (1250 cm$^{-1}$ and 816 cm$^{-1}$). For Gr-nZVI (after treatment), several peaks were lost or reduced in strength as a result of the nanocomposite adsorbing the heavy metals.

### 3.2. Batch Equilibrium Test

#### 3.2.1. Effect of the Adsorbent Dosage

The influence of the adsorbent dosage on the adsorption capacity of the adsorbents for the adsorption of heavy metals is presented in Figure S4. From the experimental data, it was revealed that the adsorption capacity decreased as the adsorbent dosage increased. For Pb (II) ions, the amount of adsorbed metal ions per gram was found to decrease from 91.88 to 0.93 mg/g (Gr-nZVI), 92.73 to 0.92 mg/g (nZVI) and 50.96 to 0.90 mg/g (Gr) with increasing adsorbent doses from 0.1 to 1.0 g. Increasing the adsorbent dosage would enhance inevitable collisions that might occur within nanoparticle adsorbents and finally lead to particle aggregation [46] and the lower adsorption capacity of the heavy metals. It was reported [47] that with the addition of a certain dose of adsorbent, the maximum adsorption was attained, and the adsorption of metal ions was found to remain constant with further increases in the dose of the adsorbent. Accordingly, for all adsorbents (Gr-nZVI, nZVI and Gr), the maximum adsorption of the metal ions was achieved at the 0.5 g/L dose application. This optimal dosage condition was subsequently used for further experiments.

#### 3.2.2. Effect of Initial Concentration

The effect of the initial metal ion concentrations of Pb(II), Cu(II), Co(II), Cd(II), Ni(II) and Zn(II) on nZVI, Gr and Gr-nZVI was investigated, and the results are shown in Figure S5. The adsorption curves of nZVI and Gr-nZVI are plotted near the x-axis for all elements. According to Wan Zuhairi and Abdul Rahim (2007) [48], a curve located near the x-axis shows a higher adsorption capacity of heavy metals. Figure S5b also shows that the removal percentage of Pb in nZVI and Gr-nZVI was almost 100%. The Pb (II) ion in Gr showed a lower adsorption capacity, ranging from $q_e$ = 1.06 to 7.77 mg/g. The Gr curve also showed that the adsorption capacity increased as the initial concentration increased until the adsorption of heavy metals became constant. Previous studies [49] indicated that the curves become constant due to the saturation of the adsorbent surface or repulsive forces between adsorbed layers and the remaining bulk molecules. At this point, no more metal ions can be adsorbed [50]. The findings agree with the removal percentage results of

all heavy metals, where, at higher equilibrium concentrations, the heavy metal removal percentage in Gr becomes lower.

Different heavy metals also show different adsorption capacities among the adsorbents. The selectivity for the removal of metal ions was as follows: nZVI (Cu > Pb > Cd > Co = Ni = Zn), Gr (Pb > Ni > Cu > Zn > Co > Cd) and Gr-nZVI (Cu > Pb > Zn > Ni > Cd > Co). For nZVI and Gr-nZVI, the probability of the Cu(II) ion forming multi-ligand complexes with the available functional groups on the adsorbent surface is higher than that for the Pb(II) ion. The Cu(II) ion's higher adsorption capacity may be due to its lower ionic radius and higher electronegativity compared to Pb (II). This would also explain the Cu(II) ion's synergistic interactive adsorption behavior in the ternary system. The lower ionic radius and higher electronegativity of the Cu(II) ion in comparison to Pb (II) may also contribute to its higher adsorption capacity. This would also explain the Cu(II) ion's synergistic interactive adsorption behavior in the multimetal system.

An adsorption isotherm expresses the relationship between the mass of heavy metals adsorbed at a constant temperature per unit mass of the sorbent and the liquid phase containing heavy metals [51]. The evaluated adsorption values, $K_d$, expressing linear, Langmuir and Freundlich isotherm models, are listed in Table S5. The $R^2$ closest to unity indicates the best fit of the isotherm to the experimental data. Adsorption studies also showed the nZVI and Gr-nZVI fitted well with the Langmuir adsorption isotherm, suggesting a homogeneous monolayer coverage. Gr and Gr-nZVI also were found to fit well with the linear equation model, as they have higher correlation values ($R^2 = 1$). However, the main disadvantage of this model is the value of the maximum adsorption capacity, and this model is limited to lower concentrations of pollutants [52,53].

Desorption or regeneration makes the adsorbent reusable, lowers secondary pollution and lowers production costs by allowing the use of a single adsorbent for multiple cycles [54,55]. This study did not consider the desorption process. However, based on previous research [54], the adsorption of Methylene Blue by a Cu(OH)2-NWs-PVA-AC nanocomposite is almost 100%, while the desorption of Methylene Blue by $H_2O$ is 2%, EtOH is 80%, NaOH is 3% and HCl is 15%. The decrease in desorption capacity was due to the increased affinity of molecules to the adsorbent with increasing adsorption capacity [54].

### 3.2.3. Effect of Contact Time

The analysis of the contact time between the adsorbent materials and heavy metal solution was conducted to investigate the minimum period required to achieve equilibrium within the adsorption system. The relationship between contact time and absorption by all adsorbent materials is shown in Figure S6. As can be seen, all curves follow the same trend, with the adsorption of metal ions onto nZVI and Gr-nZVI exhibiting linearity with time until 5 min, after which the curves were constant. In contrast, for Gr, it took 20 to 30 min to achieve linearity before the curves were constant. According to Alemayehu and Lennartz [56], the first molecules that arrive at the bare surface of an adsorbent are preferentially adsorbed on the most active sites, and the high initial uptake rate may be attributed to the availability of a large number of adsorption sites. After 5 min, the adsorption rate becomes slower, and at this point, the adsorption equilibrium has been attained. The active sites also become limited, causing the adsorption rate to decrease [49,57,58]. The ranking for heavy metal (Pb, Cu, Co, Cd, Ni and Zn) adsorption is as follows: nZVI > Gr-nZVI > Gr. According to the analysis in [41], the reduction of Cr(VI) by $Fe^0$ is cycled and involved in multiple reactions via an electrochemical corrosion mechanism.

To investigate the kinetics of the adsorption process, the experimental data were analyzed using reaction-based models (pseudo-first-order and pseudo-second-order models) and diffusion-based models (external diffusion models and internal diffusion models). All kinetic adsorption parameters of the models were calculated using the equations shown in Table S1. Based on the coefficient values of the kinetic models ($R^2$), all the adsorption studies revealed that the pseudo-second-order model fitted better ($R^2 > 0.75$) than

the pseudo-first-order model ($R^2 < 0.75$). These findings are in agreement with previous studies [59]. These findings indicate a strong interactive chemical force between metal ions and the functional groups on the nanocomposite. It also suggests a strong surface complexation between adsorbates and adsorbents [60]. Moreover, the adsorbed amount of Pb at equilibrium (17.89 mg/g) calculated with the pseudo-second-order model was close to the adsorbed amount at equilibrium obtained in the experiment (20.69 mg/g). These findings are consistent with previous studies' results [60,61].

The description of the adsorption of metal ions on the adsorbents using diffusion models is presented in Table S6. The interparticle diffusion model for all adsorbent materials showed an excellent correlation ($R^2 > 0.75$) with the adsorption data. These findings indicate that the adsorption of the heavy metals occurred on the exterior surface of the adsorbent particles. For nZVI, the selectivity of the adsorbed metal through interparticle diffusion is as follows: Cu ($K_f = 1.21$ m$^2$ g$^{-1}$ min L$^{-1}$, $R^2 = 0.89$) > Ni ($K_f = 0.87$ m$^2$ g$^{-1}$ min L$^{-1}$, $R^2 = 0.90$) > Co ($K_f = 0.80$ m$^2$ g$^{-1}$ min L$^{-1}$, $R^2 = 0.91$) > Zn ($K_f = 0.74$ m$^2$ g$^{-1}$ min L$^{-1}$, $R^2 = 0.91$) > Pb ($K_f = 0.56$ m$^2$ g$^{-1}$ min L$^{-1}$, $R^2 = 0.82$) > Cd ($K_f = 0.44$ m$^2$ g$^{-1}$ min L$^{-1}$, $R^2 = 0.84$). For intraparticle diffusion, only nZVI and Gr-nZVI fitted well with the models, while Gr mostly had a low correlation ($R^2 < 0.75$). This was due to Gr not supporting the intraparticle diffusion of heavy metals for adsorption. Comparing the adsorption values and correlation coefficients with Figure S7, the plot of the intraparticle diffusion model clearly reveals that the intraparticle diffusion of nZVI, Gr and Gr-nZVI is not the only rate-limiting step (the plots do not pass through the origin), but other kinetic models might also control the rate of adsorption, and all of them might be operating simultaneously. These results are consistent with a previous study [62]. According to [61], fast adsorption was mainly attributed to boundary layer diffusion or macro-pore diffusion, while the slow adsorption exhibited by the adsorbents was due to intraparticle diffusion or micro-pore diffusion.

### 3.2.4. Effect of pH

pH is one of the parameters affecting the heavy metal adsorption capacity, and the results are shown in Figure S8. For nZVI and Gr-nZVI, the results showed no apparent effect of pH (adsorption curves are constant) on the adsorption of heavy metals. These findings also show that nanoparticles are not sensitive to the pH value for the adsorption of heavy metals in an aqueous solution, which contrasts with the adsorption of heavy metals on Gr. However, in Figure S8e,f, the adsorption of Ni and Zn using Gr-nZVI becomes slower with increasing pH values. In a previous study (Du et al., 2014), the authors found that the oxidation rate of As(III) decreased as the pH value increased. According to [31], in acidic conditions, the surface of nZVI corrodes and produces in situ ferrous ions. This will trigger Fenton reactions when hydrogen peroxide is present (Equations (3)–(5)), resulting in the generation of OH• radicals [63]. Ferric ions can be converted to ferrous ions at the nZVI surface, which speeds up the recycling of ferric iron at the iron surface (Equation (6)); this process is called the advanced Fenton process (AFP). At higher pH (pH 8–12), the adsorption of heavy metal ions was slower than at lower pH due to the formation of ferric-hydroxo complexes, thus reducing the generation of hydroxyl radicals and finally lowering the performance of the Fenton process. These findings are also in agreement with [63]. According to [41], precipitation also occurred in the form of oxyhydroxide and metallic iron (i.e., as a reductant). Fe$^0$ nanoparticles primarily function as a reductant when removing Cr(VI). By combining with the iron oxyhydroxide shell, reduced Cr(III) can form (CrxFe1-x)(OH)3 or CrxFe1-xOOH at the surface. This structure may act as a passive layer at the surface at high initial Cr concentrations, hindering the further reduction of Cr(VI). The authors also found that the soil used as a composite contained significant amounts of minerals such as quartz, mica and ferromagnesium silicates, and these minerals may also help Fe$^0$ nanoparticles reduce Cr (VI). In their study, chromium was made less toxic.

The Gr curve shows that a higher adsorption capacity occurred at higher pH for all elements. This result is in agreement with a study [64] that reported that, in an acidic medium, a lower adsorption capacity due to the partial protonation of the active groups

tends to cause the repulsion of active groups with a positive charge [65]. However, for all elements, the heavy metal adsorption selectivity is as follows: nZVI > Gr-nZVI > Gr.

$$Fe^0 + 2 H^+ \rightarrow Fe^{2+} + H_2 \tag{3}$$

$$Fe^{2+} + H_2O_2 \rightarrow Fe^{3+} + HO\bullet + HO^- \tag{4}$$

$$HO\bullet + \text{heavy metals} \rightarrow \text{oxidize heavy metals} + H_2O \tag{5}$$

$$Fe^{3+} + Fe^0 \rightarrow 3 Fe^{2+} \tag{6}$$

### 3.2.5. Effect of Temperature

The effect of temperature on the heavy metal adsorption capacity of nZVI, Gr and Gr-nZVI is depicted in Figure S9. The experiments were carried out at temperatures ranging from 30 to 60 °C. In Figure S9, the adsorption curves of nZVI and Gr-nZVI are constant, indicating that heavy metal ions' adsorption capacity on nanoparticle adsorbents is not significantly affected by the temperature. This is a good example of an energy-efficient process because there is no need to raise the temperature of the reaction [66,67]. On the other hand, for Gr, an increased temperature led to increased adsorption capacity until, at one point (50 °C, $q_{e\ Pb}$ = 11.94 mg/g), the adsorption curves decreased. According to [68], increased temperatures will increase the diffusion rate of the adsorbate molecules across the external boundary layer. The internal pores of the adsorbent particles occurred due to a decrease in the solution's viscosity. At higher temperatures (60 °C), the adsorption of heavy metal ions decreased due to physical damage to the mineral, thus reducing its adsorption capacity [69]. This study suggested that evaluating the adsorption capacity of adsorbents at room temperature is a more sustainable process. This is because, at a higher temperature, the operational cost of the process increases, which implies that the synthesized adsorbents and the process of the adsorption of metal ions on their surfaces are less energy-intensive. Therefore, the adsorbents are economically feasible for the removal of metal ions from an aqueous solution.

### 4. Conclusions

The experimental results demonstrated that Gr-nZVI could be used as an adsorbent to adsorb heavy metals ($Pb^{2+}$, $Cu^{2+}$, $Co^{2+}$, $Cd^{2+}$, $Ni^{2+}$ and $Zn^{2+}$). From the characterization study, Gr-nZVI was successfully synthesized using the reduction method and iron $Fe^0$, which is clearly indicated by the results of XRD and FESEM. FESEM showed that Gr-nZVI was spherical-shaped, with a size ranging between 54.75 nm to 71.46 nm, and aggregated into a chain-like structure. Based on initial concentration effects, the selectivity for all heavy metals adsorbed by adsorbents is as follows: nZVI > Gr-nZVI > Gr. For the interparticle diffusion model, all adsorbents showed excellent correlations ($R^2$ > 0.75), indicating that the adsorption of all heavy metal ions was a surface process, occurring on the exterior of the adsorbent particle. It was also observed that intraparticle diffusion was not the only rate-limiting step. nZVI and Gr-nZVI also showed no apparent effects of pH and temperature (constant adsorption curves) on heavy metal adsorption. These are good signals that the nanocomposite is not sensitive to the pH value and temperature. This study suggests nZVI and Gr-nZVI are suitable for heavy metal treatment. This remediation method is highly recommended due to the high environmental efficiency of the adsorbents.

**Supplementary Materials:** The following supporting information can be downloaded at https://www.mdpi.com/article/10.3390/inorganics11030131/s1. Figure S1: The morphologies before treatment: SEM for (a) Gr (Mag: 5000×) and FESEM for (b) nZVI (Mag: 50,000×), (c) nZVI (Mag: 100,000×), (d) Gr-nZVI (Mag: 50,000×), (e) Gr-nZVI (Mag: 100,000×; the morphologies after treatment: FESEM for (f) Gr (Mag: 5000×), (g) nZVI (Mag: 50,000×), (h) Gr-nZVI (Mag: 50,000×); Figure S2: The XRD results for all samples (SB = sodium borate, $Na_2B_4O_7$; Q = quartz; K = kaolinite; H = halloysite; Fe = iron $Fe^0$); Figure S3: The FTIR spectra of nZVI, Gr and Gr-nZVI before and after treatment; Figure S4: Effect of adsorbent dosage on heavy metal ((a) Pb, (b) Cu, (c) Co, (d) Cd, (e) Ni, (f)

Zn) adsorption capacity (Co = 50 mg/L; V = 30 mL; pH = 6; shaking time = 3 h; temperature = 25 °C); Figure S5: Effects of initial concentration and removal percentage on heavy metal (Pb (a and b), Cu (c and d), Co (e and f), Cd (g and h), Ni (i and j) and Zn (k and l)) adsorption capacity (M = 0.5 g; V = 50 mL; pH = 6; shaking time = 3 h; temperature = 25 °C); Figure S6: Effect of contact time on heavy metal ((a) Pb, (b) Cu, (c) Co, (d) Cd, (e) Ni, (f) Zn) adsorption capacity (Co = 50 mg/L; M = 0.5 g; V = 50 mL; pH = 6; temperature = 25 °C); Figure S7: Plot of intraparticle diffusion modeling of heavy metals ((a) Pb, (b) Cu, (c) Co, (d) Cd, (e) Ni, (f) Zn) onto adsorbents (nZVI, Gr and Gr-nZVI); Figure S8: Effect of initial pH on heavy metal ((a) Pb, (b) Cu, (c) Co, (d) Cd, (e) Ni, (f) Zn) adsorption capacity (Co = 50 mg/L; M = 0.5 g, V = 50 mL; shaking time = 3 h; temperature = 25 °C); Figure S9: Effect of temperature on heavy metal ((a) Pb, (b) Cu, (c) Co, (d) Cd, (e) Ni, (f) Zn) adsorption capacity (Co = 50 mg/L; M = 0.5 g; V = 50 mL; pH = 6; shaking time = 3 h); Table S1: Linear equation of adsorption isotherm and kinetic model; Table S2: BET results for all samples; Table S3: CEC for all samples; Table S4: XRF results for all adsorbent materials; Table S5: Model parameters for concentration; Table S6: Kinetic parameters for heavy metal adsorption onto adsorbents.

**Author Contributions:** Conceptualization, N.'A.Z. and W.Z.W.Y.; methodology, N.'A.Z.; software, N.'A.Z.; formal analysis, N.'A.Z.; writing—original draft preparation, N.'A.Z. and A.A.O.; writing—review and editing, N.'A.Z., A.A.O. and B.S.; supervision, W.Z.W.Y., B.S., H.J. and R.C.O.; funding acquisition, B.S. and R.C.O. All authors have read and agreed to the published version of the manuscript.

**Funding:** This research was funded by Zamalah Research Scheme, Centre for Research & Instrumentation Management (CRIM), Universiti Kebangsaan Malaysia (UKM). This work also supported by Institute of Energy Infrastructure (IEI), Universiti Tenaga Nasional (UNITEN) via Postdoctoral Fellowship under grant number: J510050002-IC-6 BOLDREFRESH2025-CENTRE OF EXCELLENCE.

**Data Availability Statement:** The research data used to support the findings of this study are contained within the article and Supplementary Materials.

**Conflicts of Interest:** The authors declare no conflict of interest.

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
