# Peer review of "Effectiveness of Artificially Synthesized Granitic Residual Soil-Supported Nano Zero-Valent Iron (Gr-nZVI) as Effective Heavy Metal Contaminant Adsorbent"

_inorganics, doi:10.3390/inorganics11030131_

Round 1

Reviewer 1 Report

Article entitled Effectiveness of Artificial Synthesized Granitic Residual Soil- Supported nano Zero valent Iron (Gr-nZVI) as Effective Heavy Metals Contaminant Adsorbents written by Nur ‘Aishah Zarime, Badariah Solemon, Wan Zuhairi Wan Yaacob, Habibah Jamil, Rohayu Che Omar, Adeleke Abdulrahman Oyekanmi and submitted to Inorganics journal as deals with an important issue of heavy metals removal from wastewater.

The article is interesting and could be considered for publication in Inorganics journal. As English is not my native language, I am not able to assess language correctness. However, while reading, I found some statements missing, confusing or unclear. Below I enclose the list of my comments.

The Authors focused on the characterization of the raw material, but there is no control what happens to the material after the sorption process is applied. In many cases, the authors do not take into account the critical parameters of the process. An example would be an experiment on the influence of pH. In the acidic environment nZVI is very soluble - which suggests that the degradation of the composite begins immediately - this effect is perfectly visible in Fig 10. After 3 hours, the entire nZVI may be dissolved. The Authors, however, completely ignore this problem. It is also worth considering the influence of pH on the form of occurrence of the examined metals. Hydroxides can form in an alkaline environment - what about their solubility and how does this affect metal removal? Will it be sorption, or will it be precipitation?

Looking at the results, it seems that the difference between the composite and nZVI is small in terms of removal efficiency. What is the percentage, for example? it seems to me should have been given. If the difference is small, it makes sense to use a composite.

The Authors write that the advantage of the sorption process is the low cost of the process. However, it seems that composite synthesis is not cheap. It is worth extending the article with considerations regarding economic analysis, comparing costs with other popular sorbents. It is also worth comparing the effectiveness of sorption.

A very important issue in the application of the sorption process is the description of sorbent desorption and regeneration. Why did the Authors completely ignore this in their considerations?

The Authors decided to use very high initial concentrations of heavy metals. How does this relate to the actual concentrations in the wastewater?

The Authors say they are designing a sorbent to remove metals from wastewater. But why do they only use aqueous metal solutions? Wastewater should also be used in the research to see how a more complex matrix affects the effectiveness of metal removal. It seems that the use of wastewater should significantly reduce the metal removal efficiency.

If the authors do not have access to real wastewater, at least they should prepare model wastewater or investigate the influence of the most important ions and organic compounds on the sorption efficiency.

The introduction lacks a thorough literature review of both the use of composite sorbents, the use of ZVI and the removal of metals. This is partially done, but details are missing.

The concept of the conducted experiment is interesting and in my opinion worth developing, but the experiment performed has many shortcomings that require supplementing. Therefore, I propose to reject the article in its current form, the number of necessary corrections exceeds the standard for major revision. The experiment should be expanded and accurately described. After completing the additions, the article can be resubmited.

Author Response

Hello and Goodday the reviewer,

I have done the correction as per advised by you. And i hope you will be satisfied with this revised manuscript

Thank you 

Reviewer 2 Report

The manuscript fits the theme of the journal and has some research significance. However, the manuscript will require considerable revision before it can be accepted. 

1. The Introduction needs to be presented in paragraphs. I suggest that it be divided into at least three paragraphs.

2. How risky are the composites? It is suggested that the authors add a relevant description to the discussion.

3. The results and discussion need a lot of embellishment. The manuscript statements are not very readable and make it difficult to read and get to the point.

4. The images need to be beautified, especially Figures 2, 3, 4 and 5, and I recommend that they are processed by a drawing software.

5. Figure 7-11 Error values that need to be added to the data.

Author Response

Hello and Goodday the reviewer,

I have done the correction as per advised by you. But for the proofreading, i haven't done yet as i am planning to submit after all the reviewer satisfied the revised paper. Lastly, i hope you will be satisfied with this revised manuscript

Thank you 

Reviewer 3 Report

In this study,  Gr-nZVI was successfully synthesized by using chemical reduction of Ferric Chloride Hexahydrate, FeCl3.6H2O and Sodium Borohydride, NaBH4. The physical, chemical, mor- phology and mineralogy of all adsorbents were characterized by different characterize methods. The nZVI and Gr-nZVI also showed no significant effect on pH and temperature contributes good example of an energy. The results are interesting. However, the study lacks of novelty and some questions should be clearly stated. 

1.It would be better to divide the paragraph in “Introduction” into several paragraphs. This will be clearer and more concise.

2.nZVI is easy to oxidize. Whether is it under the nitrogen atmosphere in the synthesis process of nZVI and Gr-nZVI nanocomposite? This is very important for the performance of the material.

3.Please check the format of the manuscript. For example, “Fe2O3” in Line 202 Page 6. 

4.Figure 1, Figure 2, 3,and Figure 4 were not clear. 

5.The author studied the adsorption capacity of heavy metals (Pb2+, Cu2+, Co2+, Cd2+, Ni2+, and Zn2+) for Gr-nZVI in the single system, the adsorption capacity of Gr-nZVI in the mixed system could be investigated.

Author Response

Hello and Goodday the reviewer,

I have done the correction as per advised by you. And for the English proofreading, i will send after all the reviewer satisfied with the revised manuscript. I do really hope you will be satisfied with this revised manuscript

Thank you 

Reviewer 4 Report

Comments related to manuscript submitted to Inorganics

Line 14, it was not clear if the adsorption of these mentioned cations was done separately or no?

Line 23. The authors have mentioned “within 5 min for Pb on nZVI’’. The reviewer wants to be sure if it is Pb2+ or Pb metal? Please advise.

Lines 28 and 29, the authors have mentioned the world selectivity? The reviewer understood that the cations were adsorbed form a mixture? Please make it clear?

The authors have compared the data of nZVI and Gr-nZVI adsorbents. However, there was no comment about the pure support: granitic residual soil. Does it adsorb metal cations?

Line 41, over the years and not year [3.4]

Line 63, why the authors did not mention some references related to the use of granite waste?

The authors have to mention if the  granitic residual soil has been used as a support for any application?

Line 74, typo mistake  for controlled.

Line 73. By analogy, does the silica material has been used as support for nano zero valent iron ?

Line 85. Is it Sodium Borohidrat? Or what? Please check the correct name of the chemical?

Line 115, the authors mentioned to remove the organic compounds from the sample”? from where they will come? If it was from the granite, why they do not calcined in air at higher temperatures to remove these organic compounds before use as a support?

- The chemical  composition of the adsorbents was measured by using cation exchange capacity? The reviewer was surprised that the CEC was used to measure the chemical composition. Please could you give more details?

Line 132, in FTIR spectroscopy the term band is used instead of peak?

Line 153 I guess it is results and discussion and not “result and discussion”

Line 187; the authors have mentioned that pores surface in Gr-nZVI were predominantly filled with nZVI particles.. I guess it is not the pores on the surface, because the low value of the pore volume could be explained by the fact that the pores are originated form voids between the particles and not the pore on the surfaces? Does the SEM confirm the pore character of the surfaces?

Line 188, again the authors reported that “The pore size of Gr-nZVI also indicated the highest values (176.0732 Å) compared to other adsorbent materials. The pores are originated from the voids between the particles

Line 204, the authors stated that “The increment of iron content in nanoparticles indicates that FeO was successfully synthesized using NaBH4 and FeCl2.6H2O in both nZVI and Gr-nZVI.” Thi statement has to be confirmed with the XRD data? XRF indicates that the iron is there as iron oxide? So how the authors deduced that it was iron metal?

Lines 222-223, the intensity of 20 CPS or is very low, so how it can confirm the presence of Fe0.? The authors have described the spectra based on the position of Fe2O3. Does it mean that the samples contain iron (III) oxide?

Lines 227, the authors have mentioned that plates structure of kaolinite are detected in the SEM photos. How they are sure that it was for kaolinite phase?

Do the XRD patterns show this phase? The reviewer suggests to present the data of the XRD as the first paragraph to support their claims?

Lines 233-234; the authors claimed that As mentioned in [29], Gr consists of kaolinite which  was identified through the peak of 2.55Å and 3.55Å. The reviewer does not agree, because the major reflection of kaolinite occurs at 7 Å,

I think the authors have to check the powder XRD patterns of the pure phases? From where the sodium borate is originated?

In Figure 4, why the intensity of the Fe0 is too low and reflection is broad? Do the authors check in the literature how reflections Fe0 has?

I guess the assignments of the reflections was not properly done

The XRD pattern of Fe0 contains other reflections, they were not identified? Please advise?

Lines 272-273, “The adsorption bands at 3960 cm-1 and 3620 cm-  on Gr and Gr-nZVI samples were assigned for outer and inner hydroxyl groups respectively. The FTIR is sensitive technique, and the kaolinite OG band s are easily to be detected. I advise the authors to rewrite this sentence and to add more comments?

From where the carboxyl acid  group comes in the Fe0 sample? The reviewer checked the experimental part and it was difficult to figure out the formation of these functional groups>

Lines 278-279: the authors have mentioned a collection of ketones, quinones, carboxylic acids or esters, and the 278 C=C bonds indicated the aromatic components. From where they were originated?

Figures 6 and 7 why the authors used  heavy metals ? their study was related to cations?

Why the authors described and mentioned the data related to lead cations? How about the others

In table 5, should be cations and not element?

Lines 377-378, the authors mentioned the functional groups? From where they were originated?

Line 423,  please change does not by was not.

There are so many figures please try to attach them as supporting information

Author Response

(The authors gave the same response as above.)

Round 2

Reviewer 1 Report

This my second review of this article. The Authors responded to my comments and remarks, but this reply is far from sufficient and raises additional questions and concerns. In my review, I clearly highlighted the shortcomings in the design and performance of the experiment, as well as in discussion, pointing out what should be additionally done. I do not see the relevant corrections in the provided revised version of the manuscript. Some answers to the questions indicate serious mistakes. An example would be the reply to my comment number 3. "We cannot give in percentage because the data have been published in percentage in other journal". This raises questions where, then, the novelty of this article is and why do the Authors want to publish the same results for a second time? My other comments were almost completely ignored. The corrections made in the manuscript are only cosmetic. Since the Authors have failed to correct the manuscript, I reiterate my opinion that this manuscript should be rejected.

Author Response

Dear Reviewer,

Thank you

Reviewer 2 Report

 The manuscript has made progress after modification, but there are still two small problems. 

1. The image quality is very poor, and the fitting curve of the adsorption experiment is not shown in the figure, so it is recommended to supplement.

2. There are obvious differences in the adsorption of several heavy metal ions by Gr-nZVI synthetic materials, and the author needs to make relevant explanations.

Author Response

Dear Reviewer,

Thank you

Reviewer 4 Report

The authors have addressed the concerns of the reviewer, and they did a lot of efforts to revise their manuscript

Author Response

Dear Reviewer,

Thank you for your comments. Its really helps me to improve my writing and content.

Thank you

Round 3

Reviewer 1 Report

This is my third review of this article. The Authors answered my comments, some corrections have been made, but still control experiments are missing.

XPS-AES results after treatment are missing.

SEM results after treatment are missing.

XRD results after treatment are missing.

FTIR results after treatment are missing.

In acidic conditions ZVI may undergo corrosion and composite could be decomposed. Stability tests are essential and they are missing. 
The material was properly tested before use, but nothing is known about it after use.

Author Response

Dear Reviewer,

Thank you

Reviewer 2 Report

The manuscript has made great progress since it was revised. I suggest the author put the figure of heavy metal adsorption on the text, which is a very important parameter.

Author Response

Dear Reviewer,
